# Extreme Classification in Log Memory using Count-Min Sketch: A Case Study of Amazon Search with 50M Products

**Tharun Medini**[*]
Electrical and Computer Engineering
Rice University
Houston, TX 77005
`tharun.medini@rice.edu`

**Qixuan Huang**
Computer Science
Rice University
Houston, TX 77005
`qh5@rice.edu`

**Yiqiu Wang**
Computer Science
MIT
Cambridge, MA 02139
`yiqiuw@mit.edu`

**Vijai Mohan**
Amazon Search
Palo Alto, CA 94301
`vijaim@amazon.com`

**Anshumali Shrivastava**
Computer Science
Rice University
Houston, TX 77005
`anshumali@rice.edu`

## Abstract

In the last decade, it has been shown that many hard AI tasks, especially in NLP, can be naturally modeled as extreme classification problems leading to improved precision. However, such models are prohibitively expensive to train due to the memory blow-up in the last layer. For example, a reasonable softmax layer for the dataset of interest in this paper can easily reach well beyond 100 billion parameters (> 400 GB memory). To alleviate this problem, we present Merged-Average Classifiers via Hashing (MACH), a generic $K$-classification algorithm where memory provably scales at $O(\log K)$ without any strong assumption on the classes. MACH is subtly a count-min sketch structure in disguise, which uses universal hashing to reduce classification with a large number of classes to few embarrassingly parallel and independent classification tasks with a small (constant) number of classes. MACH naturally provides a technique for zero communication model parallelism. We experiment with 6 datasets; some multiclass and some multilabel, and show consistent improvement over respective state-of-the-art baselines. In particular, we train an end-to-end deep classifier on a private product search dataset sampled from Amazon Search Engine with 70 million queries and 49.46 million products. MACH outperforms, by a significant margin, the state-of-the-art extreme classification models deployed on commercial search engines: Parabel and dense embedding models. Our largest model has 6.4 billion parameters and trains in less than 35 hours on a single p3.16x machine. Our training times are 7-10x faster, and our memory footprints are 2-4x smaller than the best baselines. This training time is also significantly lower than the one reported by Google's mixture of experts (MoE) language model on a comparable model size and hardware.

## 1 Introduction

The area of extreme classification has gained significant interest in recent years [7, 19, 2]. In the last decade, it has been shown that many hard AI problems can be naturally modeled as massive multiclass or multilabel problems leading to a drastic improvement over prior work. For example, popular NLP models predict the best word, given the full context observed so far. Such models are becoming the state-of-the-art in machine translation [22], word embeddings [16], question answering, etc. For a large dataset, the vocabulary size can quickly run into billions [16]. Similarly, Information

---

[*]Part of this work done while interning at Amazon Search, Palo Alto, CA

Retrieval, the backbone of modern Search Engines, has increasingly seen Deep Learning models being deployed in real Recommender Systems [18, 25].

However, the scale of models required by above tasks makes it practically impossible to train a straightforward classification model, forcing us to use embedding based solutions [17, 23, 4]. Embedding models project the inputs and the classes onto a small dimensional subspace (thereby removing the intractable last layer). But they have two main bottlenecks: 1) the optimization is done on a pairwise loss function leading to a huge number of training instances (each input-class pair is a training instance) and so negative sampling [16] adds to the problem, 2) The loss functions are based on handcrafted similarity thresholds which are not well understood. Using a standard cross-entropy based classifier instead intrinsically solves these two issues while introducing new challenges.

One of the datasets used in this paper is an aggregated and sampled product search dataset from Amazon search engine which consists of 70 million queries and around 50 million products. Consider the popular and powerful p3.16x machine that has 8 V-100 GPUs each with 16GB memory. Even for this machine, the maximum number of 32-bit floating point parameters that we can have is 32 billion. If we use momentum based optimizers like Adam [14], we would need 3x parameters for training because Adam requires 2 auxiliary variables per parameter. That would technically limit our network parameter space to $\approx 10$ billion. A simplest fully connected network with a single hidden layer of 2000 nodes (needed for good accuracy) and an output space of 50 million would require $2000 \times 50 \times 10^6 = 100$ billion parameters without accounting for the input-to-hidden layer weights and the data batches required for training. Such model will need $> 1.2$ TB of memory for the parameter only with Adam Optimizer.

**Model-Parallelism requires communication:** The above requirements are unreasonable to train an end-to-end classification model even on a powerful and expensive p3.16x machine. We will need sophisticated clusters and distributed training like [20]. Training large models in distributed computing environments is a sought after topic in large scale learning. The parameters of giant models are required to be split across multiple nodes. However, this setup requires costly communication and synchronization between the parameter server and processing nodes in order to transfer the gradient and parameter updates. The sequential nature of gradient updates prohibits efficient sharding of the parameters across computer nodes. Ad-hoc model breaking is known to hurt accuracy.

**Contrast with Google's MoE model:** A notable work in large scale distributed deep learning with model parallelism is Google's 'Sparsely-Gated Mixture of Experts' [21]. Here, the authors mix smart data and model parallelism techniques to achieve fast training times for super-huge networks (with up to 137 billion parameters). One of their tasks uses the **1 billion word language modelling dataset** that has $\approx 30\ million$ unique sentences with a total of $793K$ words which is much smaller than our product search dataset. One of the configurations of 'MoE' has around **4.37 billion** parameters which is smaller than our proposed model size (we use a total of **6.4 billion**, as we will see in section 4.2). Using 32 k40 GPUs, they train 10 epochs in **47 hrs**. Our model trains 10 epochs on 8 V100 GPUs (roughly similar computing power) in just **34.2 hrs**. This signifies the impact of zero-communication distributed parallelism for training outrageously large networks, which to the best of our knowledge is only achieved by MACH.

**Our Contributions:** We propose a simple hashing based divide-and-conquer algorithm MACH (Merged-Average Classification via Hashing) for $K$-class classification, which only requires $O(d \log K)$ model size (memory) instead of $O(Kd)$ required by traditional linear classifiers ($d$ id the dimension of the penultimate layer). MACH also provides computational savings by requiring only $O(Bd \log K + K \log K)$ ($B$ is a constant that we'll see later) multiplications during inference instead of $O(Kd)$ for the last layer.

Furthermore, the training process of MACH is embarrassingly parallelizable obviating the need of any sophisticated model parallelism. We provide strong theoretical guarantees (section C in appendix) quantifying the trade-offs between computations, accuracy and memory. In particular, we show that in $\log \frac{K}{\sqrt{\delta}} d$ memory, MACH can discriminate between any two pair of classes with probability $1 - \delta$. Our analysis provides a novel treatment for approximation in classification via a distinguishability property between any pair of classes.

We do not make any strong assumptions on the classes. Our results are for any generic $K$-class classification without any relations, whatsoever, between the classes. Our idea takes the existing connections between extreme classification (sparse output) and compressed sensing, pointed out

in [12], to an another level in order to avoid storing costly sensing matrix. We comment about this in detail in section 3. Our formalism of approximate multiclass classification and its strong connections with count-min sketches [8] could be of independent interest in itself.

We experiment with multiclass datasets ODP-105K and fine-grained ImageNet-22K; multilabel datasets Wiki10-31K , Delicious-200K and Amazon-670K; and an Amazon Search Dataset with 70M queries and 50M products. MACH achieves 19.28% accuracy on the ODP dataset with 105K classes which is the best reported so far on this dataset, previous best being only 9% [9]. To achieve around 15% accuracy, the model size with MACH is merely 1.2GB compared to around 160GB with the one-vs-all classifier that gets 9% and requires high-memory servers to process heavy models. On the 50M Search dataset, MACH outperforms the best extreme multilabel classification technique Parabel by $6\%$ on weighted recall and Deep Semantic Search Model (DSSM, a dense embedding model tested online on Amazon Search) by $21\%$ (details mentioned in section 4.2.2). We corroborate the generalization of MACH by matching the best algorithms like Parabel and DisMEC on P@1, P@3 and P@5 on popular extreme classification datasets Amazon-670K, Delicious-200K and Wiki10-31K.

## 2 Background

We will use the standard [] for integer range, i.e., $[l]$ denotes the integer set from 1 to $l$: $[l] = \{1, 2, \cdots, l\}$. We will use the standard logistic regression settings for analysis. The data $D$ is given by $D = (x_i, y_i)_{i=1}^N$. $x_i \in \mathbb{R}^d$ will be $d$ dimensional features and $y_i \in \{1, 2, \cdots, K\}$, where $K$ denotes the number of classes. We will drop the subscript $i$ for simplicity whenever we are talking about a generic data point and only use $(x, y)$. Given an $x$, we will denote the probability of $y$ (label) taking the value $i$, under the given classifier model, as $p_i = Pr(y = i|x)$.

**2-Universal Hashing:** A randomized function $h : [l] \rightarrow [B]$ is 2-universal if for all, $i, j \in [l]$ with $i \neq j$, we have the following property for any $z_1, z_2 \in [k]$

$$Pr(h(i) = z_1 \text{ and } h(j) = z_2) = \frac{1}{B^2} \tag{1}$$

As shown in [5], the simplest way to create a 2-universal hashing scheme is to pick a prime number $p \geq B$, sample two random numbers $a, b$ uniformly in the range $[0, p]$ and compute $h(x) = ((ax + b) \bmod p) \bmod B$.

**Count-Min Sketch:** Count-Min Sketch is a widely used approximate counting algorithm to identify the most frequent elements in a huge stream that we do not want to store in memory.

Assume that we have a stream $a_1, a_2, a_3.....$ where there could be repetition of elements. We would like to estimate how many times each distinct element has appeared in the stream. The stream could be very long and the number of distinct elements $K$ could be large. In Count-Min Sketch [8], we basically assign $O(\log K)$ 'signatures' to each class using 2-universal hash functions. We use $O(\log K)$ different hash functions $H_1, H_2, H_3, ..., H_{O(\log K)}$, each mapping any class $i$ to a small range of buckets $B << K$, i.e., $H_j(i) \in [B]$. We maintain a counting-matrix $C$ of order $O(\log K) * B$. If we encounter class $i$ in the stream of classes, we increment the counts in cells $H_1(i), H_2(i)....., H_{O(\log K)}(i)$. It is easy to notice that there will be collisions of classes into these counting cells. Hence, the counts for a class in respective cells could be over-estimates.

During inference, we want to know the frequency of a particular element say $a_1$. We simply go to all the cells where $a_1$ is mapped to. Each cell gives and over-estimated value of the original frequency of $a_1$. To reduce the offset of estimation, the algorithm proposes to take the minimum of all the estimates as the approximate frequency, i.e., $n_{approx}(a_1) = min(C[1, H_1(i)], C[2, H_2(i)], ...., C[\log K, H_{\log K}])$. An example illustration of Count-Min Sketch is given in figure 1 in appendix.

## 3 Our Proposal: Merged-Averaged Classifiers via Hashing (MACH)

MACH randomly merges $K$ classes into $B$ random-meta-classes or buckets ($B$ is a small, manageable number). We then run any off-the-shelf classifier, such as logistic regression or a deep network, on this meta-class classification problem. We then repeat the process independently $R = O(\log K)$ times each time using an independent 2-universal hashing scheme. During prediction, we aggregate the output from each of the $R$ classifiers to obtain the predicted class. We show that this simple scheme is theoretically sound and only needs $\log K$ memory in Theorem 2. We present Information theoretic connections of our scheme with compressed sensing and heavy hitters. Figure 1 broadly explains our idea.

Formally, we use $R$, independently chosen, 2-universal hash functions $h_i : [K] \rightarrow [B]$, $i = \{1, 2, \cdots, R\}$. Each $h_i$ uniformly maps the $K$ classes into one of the $B$ buckets. $B$ and $R$ are our parameters that we can tune to trade accuracy with both computations and memory. $B$ is usually a small constant like 10 or 50. Given the data $\{x_i, y_i\}_{i=1}^N$, it is convenient to visualize that each hash function $h_j$, transforms the data $D$ into $D_j = \{x_i, h_j(y_i)\}_{i=1}^N$. We do not have to materialize the hashed class values for all small classifiers, we can simply access the class values through $h_j$. We then train $R$ classifiers, one on each of these $D_j$'s to get $R$ models $M_j$s. This concludes our training process. Note that each $h_j$ is independent. Training $R$ classifiers is trivially parallelizable across $R$ machines or GPUs.

We need a few more notations. Each meta-classifier can only classify among the merged meta-classes. Let us denote the probability of the meta-class $b \in [B]$, with the $j^{th}$ classifier with capitalized $P_b^j$. If the meta-class contains the class $i$, i.e. $h_j(i) = b$, then we can also write it as $P_{h_j(i)}^j$.

Before we describe the prediction phase, we have following theorem (proved in Section C in appendix).

**Theorem 1**

$$\mathbb{E}\left[ \frac{B}{B-1} \left[ \frac{1}{R} \sum_{j=1}^{R} P_{h_j(i)}^j(x) - \frac{1}{B} \right] \right] = Pr\left( y = i \middle| x \right) = p_i \qquad (2)$$

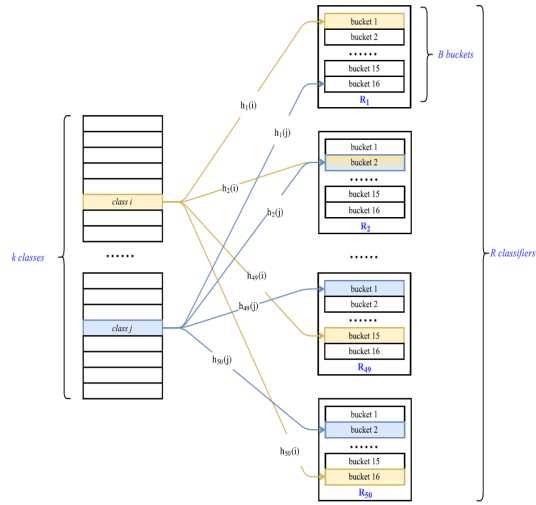

Figure 1: Outline of MACH. We hash each class into $B$ bins using a 2-universal hash function. We use $R$ different hash functions and assign different signatures to each of the $K$ classes. We then train $R$ independent $B$ class classifiers ($B << K$)

In theorem 1, $P_{h_j(i)}^j(x)$ is the predicted probability of meta-class $h_j(i)$ under the $j^{th}$ model ($M_j$), for given $x$. It's easy to observe that the true probability of a class grows linearly with the sum of individual meta-class probabilities (with multiplier of $\frac{B}{R*(B-1)}$ and a shift of $\frac{-1}{B-1}$). Thus, our classification rule is given by $\arg\max_i Pr(y = i|x) = \arg\max_i \sum_{j=1}^{R} P_{h_j(i)}^j(x)$. The pseudocode for both training and prediction phases is given in Algorithms 1 and 2 in appendix.

Clearly, the total model size of MACH is $O(RBd)$ to store $R$ models of size $Bd$ each. The prediction cost requires $RBd$ multiplications to get meta probabilities, followed by $KR$ to compute equation 1 for each of the classes. The argmax can be calculated on the fly. Thus, the total cost of prediction is $RBd + KR$. Since $R$ models are independent, both the training and prediction phases are conducive to trivial parallellization. Hence, the overall inference time can be brought down to $Bd + KR$.

To obtain significant savings on both model size and computation, we want $BR \ll K$. The subsequent discussion shows that $BR \approx O(\log K)$ is sufficient for identifying the final class with high probability.

**Definition 1 Indistinguishable Class Pairs:** *Given any two classes $c_1$ and $c_2 \in [K]$, they are indistinguishable under MACH if they fall in the same meta-class for all the $R$ hash functions, i.e., $h_j(c_1) = h_j(c_2)$ for all $j \in [R]$.*

Otherwise, there is at least one classifier which provides discriminating information between them. Given that the only sources of randomness are the independent 2-universal hash functions, we can have the following lemma:

**Lemma 1** *Using MACH with $R$ independent $B$-class classifier models, any two original classes $c_1$ and $c_2 \in [K]$ will be indistinguishable with probability at most*

$$Pr(\text{classes } i \text{ and } j \text{ are indistinguishable}) \leq \left(\frac{1}{B}\right)^R \tag{3}$$

There are total $\frac{K(K-1)}{2} \leq K^2$ possible pairs, and therefore, the probability that there exist at least one pair of classes, which is indistinguishable under MACH is given by the union bound as

$$Pr(\exists \text{ an indistinguishable pair}) \leq K^2 \left(\frac{1}{B}\right)^R \tag{4}$$

Thus, all we need is $K^2 \left(\frac{1}{B}\right)^R \leq \delta$ to ensure that there is no indistinguishable pair with probability $\geq 1 - \delta$. Overall, we get the following theorem:

**Theorem 2** *For any $B$, $R = \frac{2\log\frac{K}{\sqrt{\delta}}}{\log B}$ guarantees that all pairs of classes $c_i$ and $c_j$ are distinguishable (not indistinguishable) from each other with probability greater than $1 - \delta$.*

The extension of MACH to multilabel setting is quite straightforward as all that we need is to change softmax cross-entropy loss to binary cross-entropy loss. The training and evaluation is similar to the multiclass classification.

**Connections with Count-Min Sketch:** Given a data instance $x$, a vanilla classifier outputs the probabilities $p_i$, $i \in \{1, 2, ..., K\}$. We want to essentially compress the information of these $K$ numbers to $\log K$ measurements. In classification, the most informative quantity is the identity of $\arg\max p_i$. If we can identify a compression scheme that can recover the high probability classes from smaller measurement vector, we can train a small-classifier to map an input to these measurements instead of the big classifier.

The foremost class of models to accomplish this task are Encoder and Decoder based models like Compressive Sensing [3]. The connection between compressed sensing and extreme classification was identified in prior works [12, 10]. In [12], the idea was to use a compressed sensing matrix to compress the $K$ dimensional binary indicator vector of the class $y_i$ to a real number and solve a regression problem. While Compressive Sensing is theoretically very compelling, recovering the original predictions is done through iterative algorithms like Iteratively Re-weighted Least Squares (IRLS)[11] which are prohibitive for low-latency systems like online predictions. Moreover, the objective function to minimize in each iteration involves the measurement matrix $A$ which is by itself a huge bottleneck to have in memory and perform computations. This defeats the whole purpose of our problem since we cannot afford $O(K * \log K)$ matrix.

**Why only Count-Min Sketch? :** Imagine a set of classes $\{cats, dogs, cars, trucks\}$. Suppose we want to train a classifier that predicts a given compressed measurement of classes: $\{0.6 * p_{cars} + 0.4 * p(cats), 0.5 * p(dogs) + 0.5 * p(trucks)\}$, where $p(class)$ denotes the probability value of class. There is no easy way to predict this without training a regression model. Prior works attempt to minimize the norm between the projections of true (0/1) $K$-vector and the predicted $\log K$-vectors (like in the case of [12]). For a large $K$, errors in regression is likely to be very large.

On the other hand, imagine two meta classes $\{[cars \& trucks], [cats \& dogs]\}$. It is easier for a model to learn how to predict whether a data point belongs to '$cars \& trucks$' because the probability assigned to this meta-class is the sum of original probabilities assigned to cars and trucks. By virtue of being a union of classes, a softmax-loss function works very well. Thus, a subtle insight is that only (0/1) design matrix for compressed sensing can be made to work here. This is precisely why a CM sketch is ideal.

It should be noted that another similar alternative Count-Sketch [6] uses $[-1, 0, 1]$ design matrix. This formulation creates meta-classes of the type $[cars \& not trucks]$ which cannot be easily estimated.

## 4  Experiments

We experiment with 6 datasets whose description and statistics are shown in table 1 in appendix. The training details and P@1,3,5 on 3 multilabel datasets Wiki10-31K, Delicious-200K and Amazon-670K are also discussed in section D.3 in appendix. The brief summary of multilabel results is that

| Dataset | (B, R) | Model size Reduction | Training Time | Prediction Time per Query | Accuracy |
|---|---|---|---|---|---|
| ODP | (32, 25) | 125x | 7.2hrs | 2.85ms | 15.446% |
| Imagenet | (512, 20) | 2x | 23hrs | 8.5ms | 10.675% |

Table 1: Wall Clock Execution Times and accuracies for two runs of MACH on a single Titan X.

MACH consistently outperforms tree-based methods like FastXML [19] and PfastreXML [13] by noticeable margin. It mostly preserves the precision achieved by the best performing algorithms like Parabel [18] and DisMEC [2] and even outperforms them on half the occasions.

### 4.1 Multiclass Classification

We use the two large public benchmark datasets ODP and ImageNet from [9].

All our multiclass experiments were performed on the same server with Geforce GTX TITAN X, Intel(R) Core(TM) i7-5960X 8-core CPU @ 3.00GHz and 64GB memory. We used Tensorflow [1] to train each individual model $M_i$ and obtain the probability matrix $P_i$ from model $M_i$. We use OpenCL to compute the global score matrix that encodes the score for all classes $[1, K]$ in testing data and perform argmax to find the predicted class. Our codes and scripts are hosted at the repository `https://github.com/Tharun24/MACH/`.

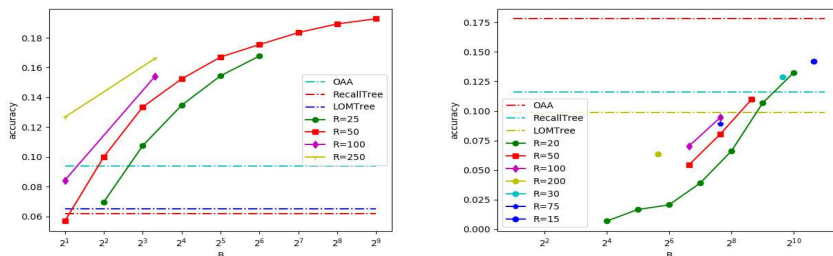

Figure 2: Accuracy Resource tradeoff with MACH (bold lines) for various settings of $R$ and $B$. The number of parameters are $BRd$ while the prediction time requires $KR + BRd$ operations. All the runs of MACH requires less memory than OAA. The straight line are accuracies of OAA, LOMTree and Recall Tree (dotted lines) on the same partition taken from [9]. LOMTree and Recall Tree uses more (around twice) the memory required by OAA. **Left:** ODP Dataset. **Right:** Imagenet Dataset

#### 4.1.1 Accuracy Baselines

On these large benchmarks, there are three published methods that have reported successful evaluations – 1) **OAA**, traditional one-vs-all classifiers, 2) **LOMTree** and 3) **Recall Tree**. The results of all these methods are taken from [9]. OAA is the standard one-vs-all classifiers whereas LOMTree and Recall Tree are tree-based methods to reduce the computational cost of prediction at the cost of increased model size. Recall Tree uses twice as much model size compared to OAA. Even LOMtree has significantly more parameters than OAA. Thus, our proposal MACH is the only method that reduces the model size compared to OAA.

#### 4.1.2 Results and Discussions

We run MACH on these two datasets varying $B$ and $R$. We used plain cross entropy loss without any regularization. We plot the accuracy as a function of different values of $B$ and $R$ in Figure 2. We use the unbiased estimator given by Equation 1 for inference as it is superior to other estimators (See section D.2 in appendix for comparison with min and median estimators).

The plots show that for ODP dataset MACH can even surpass OAA achieving 18% accuracy while the best-known accuracy on this partition is only 9%. LOMtree and Recall Tree can only achieve 6-6.5% accuracy. It should be noted that with 100,000 classes, a random accuracy is $10^{-5}$. Thus, the improvements are staggering with MACH. Even with $B = 32$ and $R = 25$, we can obtain more than 15% accuracy with $\frac{105,000}{32 \times 25} = 120$ times reduction in the model size. Thus, OAA needs 160GB model size, while we only need around 1.2GB. To get the same accuracy as OAA, we only need $R = 50$ and $B = 4$, which is a 480x reduction in model size requiring mere 0.3GB model file.

On ImageNet dataset, MACH can achieve around 11% which is roughly the same accuracy of LOMTree and Recall Tree while using $R = 20$ and $B = 512$. With $R = 20$ and $B = 512$, the

memory requirement is $\frac{21841}{512 \times 20} = 2$ times less than that of OAA. On the contrary, Recall Tree and LOMTree use 2x more memory than OAA. OAA achieves the best result of $17\%$. With MACH, we can run at any memory budget.

In table 1, we have compiled the running time of some of the reasonable combination and have shown the training and prediction time. The prediction time includes the work of computing probabilities of meta-classes followed by sequential aggregation of probabilities and finding the class with the max probability. The wall clock times are significantly faster than the one reported by RecallTree, which is optimized for inference.

## 4.2 Information Retrieval with 50 million Products

After corroborating MACH's applicability on large public extreme classification datasets, we move on to the much more challenging real Information Retrieval dataset with 50M classes to showcase the power of MACH at scale. As mentioned earlier, we use an aggregated and sub-sampled search dataset mapping queries to product purchases. Sampling statistics are hidden to respect Amazon's disclosure policies.

The dataset has 70.3 M unique queries and 49.46 M products. For every query, there is atleast one purchase from the set of products. Purchases have been amalgamated from multiple categories and then uniformly sampled. The average number of products purchased per query is 2.1 and the average number of unique queries per product is 14.69.

For evaluation, we curate another 20000 unique queries with atleast one purchase among the afore-mentioned product set. These transactions sampled for evaluation come from a time-period that succeeds the duration of the training data. Hence, there is no temporal overlap between the transactions. Our goal is to measure whether our top predictions contain the true purchased products, i.e., we are interested in measuring the purchase recall.

For measuring the performance on Ranking, for each of the 20000 queries, we append 'seen but not purchased' products along with purchased products. To be precise, every query in the evaluation dataset has a list of products few of which have been purchased and few others that were clicked but not purchased (called 'seen negatives'). On an average, each of the 20000 queries has $14$ products of which $\approx 2$ are purchased and the rest are 'seen negatives'. Since products that are only 'seen' are also relevant to the query, it becomes challenging for a model to selectively rank purchases higher than another related products that were not purchased. A good model should be able to identify these subtle nuances and rank purchases higher than just 'seen' ones.

Each query in the dataset has sparse feature representation of 715K comprising of 125K frequent word unigrams, 20K frequent bigrams and 70K character trigrams and 500K reserved slots for hashing out-of-vocabulary tokens.

**Architecture** Since MACH fundamentally trains many small models, an input dimension of 715K is too large. Hence, we use sklearn's $murmurhash3\_32$ package and perform feature hashing [24] to reduce the input dimension to 80K (empirically observed to have less information loss). We use a feed forward neural network with the architecture 80K-2K-2K-$B$ for each of $R$ classifiers. 2000 is the embedding dimension for a query, another 2000 is the hidden layer dimension and the final output layer is $B$ dimensional where we report the metrics with $B = 10000$ and $B = 20000$. For each $B$, we train a maximum of 32 repetitions ,i.e., $R = 32$. We show the performance trend as $R$ goes from 2,4,8,16,32.

**Metrics** Although we pose the Search problem as a multilabel classification model, the usual precision metric is not enough to have a clear picture. In product retrieval, we have a multitude of metrics in consideration (all metrics of interest are explained in section E in appendix). Our primary metric is weighted Recall@100 (we get the top 100 predictions from our model and measure the recall) where the weights come from number of sessions. To be precise, if a query appeared in a lot of sessions, we prioritize the recall on those queries as opposed to queries that are infrequent/unpopular. For example, if we only have 2 queries $q_1$ and $q_2$ which appeared in $n_1$ and $n_2$ sessions respectively. The $wRecall@100$ is given by $(recall@100(q_1) * n_1 + recall@100(q_2) * n_2)/(n_1 + n_2)$. Un-weighted recall is the simple mean of Recall@100 of all the queries.

### 4.2.1 Baselines

**Parabel:** A natural comparison would arise with the recent algorithm 'Parabel' [18] as it has been used in Bing Search Engine to solve a 7M class challenge. We compare our approach with the publicly available code of Parabel. We vary the number of trees among 2,4,8,16 and chose the maximum

number of products per leaf node to vary among 100, 1000 and 8000. We have experimented with a few configurations of Parabel and figured out that the configuration with 16 trees each with 16300 nodes (setting the maximum number of classes in each leaf node to 8000) gives the best performance. In principle, number of trees in Parabel can be perceived as number of repetitions in MACH. Similarly, number of nodes in each tree in Parabel is equivalent to number of buckets $B$ in MACH. We could not go beyond 16 trees in Parabel as the memory consumption was beyond limits (see Table 2).

**Deep Semantic Search Model (DSSM)**: We tried running the publicly available C++ code of AnnexML [23] (graph embedding based model) by varying embedding dimension and number of learners. But none of the configurations could show any progress even after 5 days of training. The next best embedding model SLEEC [4] has a public MATLAB code but it doesn't scale beyond 1 million classes (as shown in extreme classification repository [15]).

In the wake of these scalability challenges, we chose to compare against a dense embedding model DSSM [17] that was A/B tested online on Amazon Search Engine. This custom model learns an embedding matrix that has a 256 dimensional dense vectors for each token (tokenized into word unigrams, word bigrams, character trigrams as mentioned earlier). This embedding matrix is shared across both queries and products. Given a query, we first tokenize it, perform a sparse embedding lookup from the embedding matrix and average the vectors to yield a vector representation. Similarly, given a product (in our case, we use the title of a product), we tokenize it and perform sparse embedding lookup and average the retrieved vectors to get a dense representation. For every query, purchased products are deemed to be highly relevant. These product vectors are supposed to be 'close' to the corresponding query vectors (imposed by a loss function). In addition to purchased products, 6x number of random products are sampled per query. These random products are deemed irrelevant by a suitable loss function.

**Objective Function**: All the vectors are unit normalized and the cosine similarity between two vectors is optimized. For a query-product pair that is purchased, the objective function enforces the cosine similarity to be $> \theta_p$ ($p$ for purchased). For a query-product pair that's deemed to be irrelevant, the cosine similarity is enforced to be $< \theta_r$ ($r$ for random).

Given the cosine similarity $s$ between a query-document pair and a label $l$ (indicating $p, r$), the overall loss function is given as $loss(s, l) = I^p(l) * min^2(0, s - \theta_p) + I^r(l) * max^2(0, s - \theta_r)$ for the embedding model. The thresholds used during online testing were $\theta_p = 0.9$ and $\theta_r = 0.1$.

### 4.2.2 Results

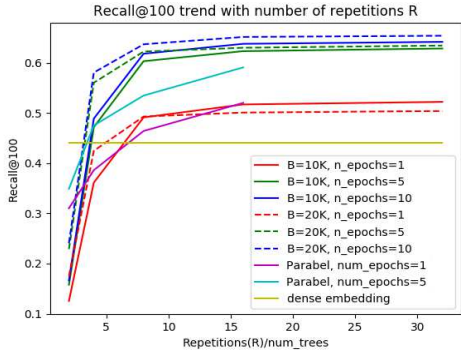

Figure 3: Comparison of our proposal MACH to Parabel and Embedding model.

Figure 3 shows the recall@100 for R=2,4,8,16,32 after 1,5 and 10 epochs respectively. The dotted red/blue/green lines correspond to MACH with $B = 20K$ and the solid red/blue/green lines correspond to $B = 10K$. The cyan and magenta lines correspond to Parabel algorithm with the number of trees being 2,4,8,16 end epochs being 1 and 5 respectively. We couldn't go beyond 16 trees for Parabel because the peak memory consumption during both training and testing was reaching the limits of the AWS p3.16x machine that we used (64 vcpus, 8 V-100 NVIDIA Tesla GPUs, 480 GB Memory). The yellow line corresponds to the dense embedding based model. The training time and memory consumption is given in table 2.

Since calculating all Matching and Ranking metrics with the entire product set of size 49.46 M is cumbersome, we come up with a representative comparison by limiting the products to just 1 M. Across all the 20000 queries in the evaluation dataset, there are 32977 unique purchases. We first retain these products and then sample the remaining 967023 products randomly from the 49.46 M products. Then we use our model and the baseline models to obtain a 20K*1 M score matrix. We then evaluate all the metrics on this sub-sampled representative score matrix. Tables 3 and 4 show the comparison of various metrics for MACH vs Parabel vs Embedding model.

| Model | epochs | wRecall | Total training time | Memory(Train) | Memory (Eval) | #Params |
|---|---|---|---|---|---|---|
| DSSM, 256 dim | 5 | 0.441 | 316.6 hrs | **40 GB** | 286 GB | 200 M |
| Parabel, num_trees=16 | 5 | 0.5810 | 232.4 hrs (all 16 trees in parallel) | 350 GB | 426 GB | - |
| MACH, B=10K, R=32 | 10 | **0.6419** | **31.8 hrs** (all 32 repetitions in parallel) | 150 GB | **80 GB** | 5.77 B |
| MACH, B=20K, R=32 | 10 | **0.6541** | **34.2 hrs** (all 32 repetitions in parallel) | 180 GB | 90 GB | 6.4 B |

Table 2: Comparison of the primary metric weighted_Recall@100, training time and peak memory consumption of MACH vs Parabel vs Embedding Model. We could only train 16 trees for Parabel as we reached our memory limits

| Metric | Embedding | Parabel | MACH, B=10K, R=32 | MACH, B=20K, R=32 |
|---|---|---|---|---|
| map_weighted | 0.6419 | 0.6335 | 0.6864 | **0.7081** |
| map_unweighted | 0.4802 | **0.5210** | 0.4913 | 0.5182 |
| mrr_weighted | 0.4439 | **0.5596** | 0.5393 | 0.5307 |
| mrr_unweighted | 0.4658 | **0.5066** | 0.4765 | 0.5015 |
| ndcg_weighted | 0.7792 | 0.7567 | 0.7211 | **0.7830** |
| ndcg_unweighted | 0.5925 | 0.6058 | 0.5828 | **0.6081** |
| recall_weighted | 0.8391 | 0.7509 | 0.8344 | **0.8486** |
| recall_unweighted | **0.8968** | 0.7717 | 0.7883 | 0.8206 |

Table 3: Comparison of Matching metrics for MACH vs Parabel vs Embedding Model. These metrics are for representative 1M products as explained.

| Metric | Embedding | Parabel | MACH, B=10K, R=32 | MACH, B=20K, R=32 |
|---|---|---|---|---|
| ndcg_weighted | 0.7456 | 0.7374 | **0.7769** | 0.7749 |
| ndcg_unweighted | 0.6076 | **0.6167** | 0.6072 | 0.6144 |
| mrr_weighted | 0.9196 | 0.9180 | 0.9414 | **0.9419** |
| mrr_unweighted | 0.516 | 0.5200 | 0.5200 | **0.5293** |
| mrr_most_rel_weighted | 0.5091 | 0.5037 | **0.5146** | 0.5108 |
| mrr_most_rel_unweighted | 0.4671 | 0.4693 | 0.4681 | **0.4767** |
| prec@1_weighted | 0.8744 | 0.8788 | **0.9109** | 0.9102 |
| prec@1_unweighted | 0.3521 | 0.3573 | 0.3667 | **0.3702** |
| prec@1_most_rel_weighted | 0.3776 | 0.3741 | 0.3989 | **0.3989** |
| prec@1_most_rel_unweighted | 0.3246 | 0.3221 | 0.3365 | **0.3460** |

Table 4: Comparison of Ranking metrics. These metrics are for curated dataset where each query has purchases and 'seen negatives' as explained in 4.2. We rank purchases higher than 'seen negatives'.

### 4.2.3 Analysis

MACH achieves considerably superior wRecall@100 compared to Parabel and the embedding model (table 2). MACH's training time is 7x smaller than Parabel and 10x smaller than embedding model for the same number of epochs. This is expected because Parabel has a partial tree structure which cannot make use of GPUs like MACH. And the embedding model trains point wise loss for every query-product pair unlike MACH which trains a multilabel cross-entropy loss per query. Since the query-product pairs are huge, the training time is very high. Memory footprint while training is considerably low for embedding model because its training just an embedding matrix. But during evaluation, the same embedding model has to load all 256 dimensional vectors for products in memory for a nearest neighbour based lookup. This causes the memory consumption to grow a lot (this is more concerning if we have limited GPUs). Parabel has high memory consumption both while training and evaluation.

We also note that MACH consistently outperforms other two algorithms on Matching and Ranking metrics (tables 3 and 4). Parabel seems to be better on MRR for matching but the all important recall is much lower than MACH.

**Acknowledgments**

The work was supported by NSF-1652131, NSF-BIGDATA 1838177, AFOSR-YIPFA9550-18-1-0152, Amazon Research Award, and ONR BRC grant for Randomized Numerical Linear Algebra.

We thank Priyanka Nigam from Amazon Search for help with data pre-processing, running the embedding model baseline and getting Matching and Ranking Metrics. We also thank Choon-Hui Teo and SVN Vishwanathan for insightful discussions about MACH's connections to different Extreme Classification paradigms.

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
