[Supplementary Material]

# Extreme Classification in Log Memory using Count-Min Sketch: A Case Study of Amazon Search with 50M Products

**Tharun Medini**[*]
Electrical and Computer Engineering
Rice University
Houston, TX 77005
tharun.medini@rice.edu

**Qixuan Huang**
Computer Science
Rice University
Houston, TX 77005
qh5@rice.edu

**Yiqiu Wang**
Computer Science
MIT
Cambridge, MA 02139
yiqiuw@mit.edu

**Vijai Mohan**
Amazon Search
Palo Alto, CA 94301
vijaim@amazon.com

**Anshumali Shrivastava**
Computer Science
Rice University
Houston, TX 77005
anshumali@rice.edu

# Appendix:MACH

## A    Count-Min Sketch

Count-Min Sketch is a widely used approximate counting algorithm to identify the most frequent elements in a huge stream that we do not want to store in memory. An example illustration of Count-Min Sketch is given in figure 1.

|   | H1 | H2 | H3 | H4 |
|---|----|----|----|----|
| A | 1  | 6  | 3  | 1  |
| B | 1  | 2  | 4  | 6  |
| C | 3  | 4  | 1  | 6  |
| D | 6  | 2  | 4  | 1  |

|    | 0 | 1 | 2 | 3 | 4 | 5 | 6 |
|----|---|---|---|---|---|---|---|
| H1 | 0 | 1+1+1+1=4 |   | 1+1 = 2 | 0 | 0 | 1 |
| H2 | 0 | 0 | 1+1=2 | 0 | 1+1 = 2 | 0 | 1+1+1=3 |
| H3 | 0 | 1+1 = 2 | 0 | 1+1+1=3 | 1+1 = 2 | 0 | 0 |
| H4 | 0 | 1+1+1+1=4 | 0 | 0 | 0 | 0 | 1+1+1=3 |

Figure 1: Illustration of count-min sketch for a stream of letters ABCAACD. The hash codes for each letter for 4 different hash functions is shown on the left and the accumulated counts for each of the letter in the stream is shown on the right

---

[*]Part of this work done while interning at Amazon Search, Palo Alto, CA

## B  Pseudocode

**Algorithm 1** Train

1: **Data:** $D = (X, Y) = (x_i,\ y_i)_{i=1}^{N}$. $x_i \in \mathbb{R}^d$ $y_i \in \{1, 2, \cdots, K\}$
2: **Input:** $B, R$
3: **Output:** $R$ trained classifiers
4: initialize $R$ 2-universal hash functions $h_1, h_2, ...h_R$
5: initialize $result$ as an empty list

6: **for** $i = 1 : R$ **do**
7:    $Y_{h_i} \leftarrow h_i(Y)$
8:    $M_i = trainClassifier(X, Y_{h_i})$
9:    Append $M_i$ to $result$
10: **end for**
11: **Return** result

**Algorithm 2** Predict

1: **Data:** $D = (X, Y) = (x_i,\ y_i)_{i=1}^{N}$. $x_i \in \mathbb{R}^d$ $y_i \in \{1, 2, \cdots, K\}$
2: **Input:** $M = M_1, M_2, ..., M_R$
3: **Output:** $N$ predicted labels
4: load $R$ 2-universal hash functions $h_1, h_2, ...h_R$ used in training
5: initialize $P$ as a an empty list
6: initialize $G$ as a $(|N| * K)$ matrix
7: **for** $i = 1 : R$ **do**
8:    $P_i = getProbability(X, M_i)$
9:    Append $P_i$ to $P$
10: **end for**
11: **for** $j = 1 : K$ **do**
12:    /* $G[:, j]$ indicates the $j$th column in matrix $G$ */
13:    $G[:, j] = (\sum_{r=1}^{R} P_r[:, h_r(j)])/R$
14: **end for**
15: **Return:** argmax(G, axis=1)

## C  Theoretical Analysis

We begin with re-emphasizing that we do not make any assumption on the classes, and we do not assume any dependencies between them. As noted before, we use $R$ independent $B$-class classifiers each. Classification algorithms such as logistic regression and deep networks models the probability $Pr(y = i|x) = p_i$. For example, the famous softmax or logistic modelling uses $Pr(y = i|x) = \frac{e^{\theta_i \cdot x}}{Z}$, where $Z$ is the partition function. With MACH, we use $R$ 2-universal hash functions. For every hash function $j$, we instead model $Pr(y = b|x) = P_b^j$, where $b \in [B]$. Since $b$ is a meta-class, we can also write $P_b^j$ as

$$P_b^j = \sum_{i:h_j(i)=b} p_i; \qquad 1 = \sum_{i=1}^{K} p_i = \sum_{b \in [B]} P_b^j \quad \forall j \tag{1}$$

With the above equation, given the $R$ classifier models, an unbiased estimator of $p_i$ is:

**Theorem 1:**

$$\mathbb{E}\left[ \frac{B}{B-1} \left[ \frac{1}{R} \sum_{j=1}^{R} P_{h_j(i)}^j(x) - \frac{1}{B} \right] \right] = Pr\left( y = i \Big| x \right) = p_i$$

**Proof :** Since the hash function is universal, we can always write

$$P_{h(i)}^j = p_i + \sum_{k \neq i} \mathbf{1}_{h(k)=h(i)} p_k,$$

where $\mathbf{1}_{h(k)=h(i)}$ is an indicator random variable with expected value of $\frac{1}{B}$. Thus $E(P_{h(i)}^j) = p_i + \frac{1}{B} \sum_{k \neq i} p_k = p_i + (1 - p_i)\frac{1}{B}$. This is because the expression $\sum_{k \neq i} p_k = 1 - p_i$ as the total probability sum up to one (assuming we are using logistic type classifiers). Simplifying, we get $p_i = \frac{B}{B-1}(E(P_{h(i)}^j)(x) - \frac{1}{B})$. It is not difficult to see that this value is also equal to $\mathbb{E}\left[ \frac{B}{B-1} \left[ \frac{1}{R} \sum_{j=1}^{R} P_{h_j(i)}^j(x) - \frac{1}{B} \right] \right]$ using linearity of expectation and the fact that $E(P_{h(i)}^j) = E(P_{h(i)}^k)$ for any $j \neq k$.

**Definition 1 Indistinguishable Class Pairs:** *Given any two classes $c_1$ and $c_2 \in [K]$, they are indistinguishable under MACH if they fall in the same meta-class for all the $R$ hash functions, i.e., $h_j(c_1) = h_j(c_2)$ for all $j \in [R]$.*

Otherwise, there is at least one classifier which provides discriminating information between them. Given that the only sources of randomness are the independent 2-universal hash functions, we can have the following lemma:

**Lemma 1** *MACH with $R$ independent $B$-class classifier models, any two original classes $c_1$ and $c_2$ $\in [K]$ will be indistinguishable with probability at most*

$$Pr(\text{classes } i \text{ and } j \text{ are indistinguishable}) \leq \left(\frac{1}{B}\right)^R \tag{2}$$

There are total $\frac{K(K-1)}{2} \leq K^2$ possible pairs, and therefore, the probability that there exist at least one pair of classes, which is indistinguishable under MACH is given by the union bound as

$$Pr(\exists \text{ an indistinguishable pair}) \leq K^2 \left(\frac{1}{B}\right)^R \tag{3}$$

Thus, all we need is $K^2 \left(\frac{1}{B}\right)^R \leq \delta$ to ensure that there is no indistinguishable pair with probability $\geq 1 - \delta$. Overall, we get the following theorem:

**Theorem 2:** For any $B$, $R = \frac{2 \log \frac{K}{\sqrt{\delta}}}{\log B}$ guarantees that all pairs of classes $c_i$ and $c_j$ are distinguishable (not indistinguishable) from each other with probability greater than $1 - \delta$.

Our memory cost is $BRd$ to guarantee all pair distinguishably with probability $1 - \delta$, which is equal to $\frac{2 \log \frac{K}{\sqrt{\delta}}}{\log B} Bd$. This holds for any constant value of $B \geq 2$. Thus, we bring the dependency on memory from $O(Kd)$ to $O(\log Kd)$ in general with approximations. Our inference cost is $\frac{2 \log \frac{K}{\sqrt{\delta}}}{\log B} Bd + \frac{2 \log \frac{K}{\sqrt{\delta}}}{\log B} K$ which is $O(K \log K + d \log K)$, which for high dimensional dataset can be significantly smaller than $Kd$.

## C.1 Subtlety of MACH

The measurements in Compressive Sensing are not a probability distribution but rather a few linear combinations of original probabilities. Imagine a set of classes $\{cats, dogs, cars, trucks\}$. Suppose we want to train a classifier that predicts a compressed distribution of classes like $\{0.6 * cars + 0.4 * cats, \ 0.5 * dogs + 0.5 * trucks\}$. There is no intuitive sense to these classes and we cannot train a model using softmax-loss which has been proven to work the best for classification. We can only attempt to train a regression model to minimize the norm(like $L_1$-norm or $L_2$-norm) between the projections of true $K$-vector and the predicted $K$-vectors(like in the case of [12]). This severely hampers the learnability of the model as classification is more structured than regression. On the other hand, imagine two conglomerate or meta classes $\{[cars \ and \ trucks], [cats \ and \ dogs]\}$. It is easier for a model to learn how to predict whether a data point belongs to '$cars \ and \ trucks$' because the probability assigned to this meta-class is the sum of original probabilities assigned to cars and trucks. By virtue of being a union of classes, a softmax-loss function would work very well unlike the case of Compressive Sensing.

This motivates us to look for counting based algorithms with frequencies of grouped classes. We can pose our problem of computing a $\log K$-vector that has information about all $K$ probabilities as the challenge of computing the histogram of $K$-classes as if they were appearing in a stream where at each time, we pick a class $i$ independently with probability $p_i$. This is precisely the classic Heavy-Hitters problem[? ].

If we assume that $\max p_i \geq \frac{1}{m}$, for sufficiently small $m \leq K$, which should be true for any good classifier. We want to identify $\arg \max p_i$ with $\sum p_i = 1$ and $\max p_i \geq \frac{1}{m} \sum p_i$ by storing sub-linear information.

Count-Min Sketch [8] is the most popular algorithm for solving this heavy hitters problem over positive streams. Please refer to section 2 in main paper (and section A in appendix) for an explanation of Count-Min Sketch. Our method of using $R$ universal hashes with $B$ range is precisely the normalized count-min sketch measurements, which we know preserve sufficient information to

identify heavy hitters (or sparsity) under good signal-to-noise ratio. Thus, if $\max p_i$ (signal) is larger than $p_j$, $j \neq i$ (noise), then we should be able to identify the heavy co-ordinates (sparsity structure) in $sparsity \times \log K$ measurements (or memory) [**?** ].

## D   Experiments

### D.1   Datasets

| Name | Type | #Train | #Test | #Classes | #Features |
|------|------|--------|-------|----------|-----------|
| ODP | Text | 1084404 | 493014 | 105033 | 422713 |
| Fine-grained Imagenet | Images | 12777062 | 1419674 | 21841 | 6144 |
| Wiki10-31K | Text | 14146 | 6616 | 30938 | 101938 |
| Delicious-200K | Text/Social Networks | 196606 | 100095 | 205443 | 782585 |
| Amazon-670K | Recommendations | 490449 | 153025 | 670091 | 135909 |
| Amazon Search Dataset | Information Retrieval | 70301491 | 20000 | 49462358 | 715000 |

Table 1: Statistics of all 6 datasets. First 2 are multiclass, next 3 are multilabel and the last is a real Search Dataset

**1) ODP:** ODP is a multiclass dataset extracted from Open Directory Project, the largest, most comprehensive human-edited directory of the Web. Each sample in the dataset is a document, and the feature representation is bag-of-words. The class label is the category associated with the document. The dataset is obtained from [7]. The input dimension $d$, number of classes $K$, training samples and testing samples are 422713, 105033, 1084404 and 493014 respectively.

**2) Fine-Grained ImageNet:** ImageNet is a dataset consisting of features extracted from an intermediate layer of a convolutional neural network trained on the ILVSRC2012 challenge dataset. Please see [7] for more details. The class label is the fine-grained object category present in the image. The input dimension $d$, number of classes $K$, training samples and testing samples are 6144, 21841, 12777062 and 1419674 respectively.

**3) Delicious-200K:** Delicious-200K dataset is a sub-sampled dataset generated from a vast corpus of almost 150 million bookmarks from Social Bookmarking Systems, del.icio.us. The corpus records all the bookmarks along with a description, provided by users (default as the title of the website), an extended description and tags they consider related.

**4) Amazon-670K:** Amazon-670K dataset is a product recommendation dataset with 670K labels. Here, each input is a vector representation of a product, and the corresponding labels are other products (among 670K choices) that a user might be interested in purchase. This is an anonymized and aggregated behavior data from Amazon and poses a significant challenge owing to a large number of classes.

### D.2   Effect of Different Estimators

| Dataset | Unbiased | Min | Median |
|---------|----------|-----|--------|
| ODP | **15.446** | 12.212 | 14.434 |
| Imagenet | 10.675 | 9.743 | **10.713** |

Table 2: Classification accuracy with three different estimators from sketches (see section D.2 for details). The training configuration are given in Table 2 in main paper

Once we have identified that our algorithm is essentially count-min sketch in disguise, it naturally opens up two other possible estimators, in addition to Equation **??** for $p_i$. The popular min estimator, which is used in traditional count-min estimator given by:

$$\hat{p_i}^{min} = \min_j P^j_{h_j(i)}(x).$$
(4)

We can also use the median estimator used by count-median sketch [6] which is another popular estimator from the data streams literature:

$$\hat{p_i}^{med} = \underset{j}{\text{median}} \, P^j_{h_j(i)}(x).$$

(5)

The evaluation with these two estimators, the min and the median, is shown in table 2. We use the same trained multiclass model from main paper and use three different estimators, the original mean estimator in main paper along with Eqn.s 4 and 5 respectively, for estimating the probability. The estimation is followed by argmax to infer the class label. It turns out that our unbiased estimator shown in main paper performs overall the best. Median is slightly better on ImageNet data and poor on ODP dataset. Min estimator leads to poor results on both of them.

### D.3 Multilabel Classification

In this section, we show that MACH preserves the fundamental metrics precision@1,3,5 (denoted hereafter by P@1, P@3 and P@5) on 3 extreme classification datasets available at XML Repository [**?**]. We chose Wiki10-31K, Delicious-200K and Amazon-670K with 31K, 200K and 670K classes respectively. This choice represents good variation in the number of classes as well as in the sparsity of labels. This section is more of a sanity check that MACH is comparable to state-of-the-art methodologies on public datasets, where memory is not critical.

The detailed comparison of P@k with state-of-the-art algorithms is given in the table 3. We notice that MACH consistently outperforms tree-based methods like FastXML [19] and PfastreXML [13] by noticeable margin. It mostly preserves the precision achieved by the best performing algorithms like Parabel [18] and DisMEC [2] and even outperforms them on few occasions. For all the baselines, we use the reported metrics (on XML Repository) on these datasets. We use the same train/test split for MACH as other baseline algorithms.

We fixed $R = 32$ and experimented with a few limited configurations of $B$. The input dimension $d$ and classes $K$ is given below the respective dataset name in table 3. The network architecture takes the form $d$-500-500-$B$. $B$ was varied among $\{1000, 2000\}$ for Wiki10-31K, among $1000, 5000$ for Delicious-200K and among $\{5000, 10000\}$ for Amazon-670K. The training and evaluation details are as follows:

**Wiki10-31K**: The network architecture we used was 101938-500-500-$B$ where $B \in \{1000, 2000\}$. Here, 101938 is the input sparse feature dimension, and we have two hidden layers of 500 nodes each. The reported result is for $B = 2000$. For $B = 1000$, there a marginal drop in precision (P@1 of $84.74\%$ vs $85.44\%$) which is expected. We trained for a total of 60 epochs with each epoch taking 20.45 seconds. All 32 repetitions were trained in parallel. Hence, total training time is 1227s. The evaluation time is 2.943 ms per test sample.

**Delicious-200K**: The network architecture we used was 782585-500-500-$B$ where $B \in \{1000, 5000\}$. Here, 782585 is the input sparse feature dimension, and we have two hidden layers of 500 nodes each. The reported result is for $B = 5000$. Remarkably, $B = 1000$ was performing very similar to $B = 5000$ in terms of precision. We trained for a total of 20 epochs with each epoch taking 187.2 seconds. 8 repetitions were trained at a time in parallel. Hence, we needed 4 rounds of training to train $R = 32$ repetitions and the total training time is 4*20*187.2 = 14976 seconds (4.16 hrs). The evaluation time is 6.8 ms per test sample.

**Amazon-670K**: The network architecture we used was 135909-500-500-$B$ where $B \in \{5000, 10000\}$. Here, 135909 is the input sparse feature dimension and we have two hidden layers of 500 nodes each. The reported result is for $B = 10000$. For $B = 5000$, there a marginal drop in precision (P@1 of $41.41\%$ vs $40.64\%$) which is expected. We trained for a total of 40 epochs with each epoch taking 132.7 seconds. All 32 repetitions were trained in parallel. Hence, total training time is 5488s (1.524 hrs). The evaluation time is 24.8 ms per test sample.

In all the above cases, we observed the P@1,3,5 for $R = 2, 4, 8, 16, 32$. We noticed that the increment in precision from $R = 16$ to $R = 32$ is very minimal. This is suggestive of saturation and hence our choice of $R = 32$ is justified. From these results and impressions, it is clear that MACH is a very viable and robust extreme classification algorithm that can scale to a large number of classes.

| Dataset | P@k | MACH | PfastreXML | FastXML | Parabel | DisMEC |
|---|---|---|---|---|---|---|
| Wiki10-31K | P@1 ($B = 2000$) | **0.8544** | 0.8357 | 0.8303 | 0.8431 | 0.8520 |
| $d = 101938$ | P@3 ($B = 2000$) | 0.7142 | 0.6861 | 0.6747 | 0.7257 | **0.7460** |
| $K = 30938$ | P@5 ($B = 2000$) | 0.6151 | 0.5910 | 0.5776 | 0.6339 | **0.6590** |
| Delicious-200K | P@1 ($B = 5000$) | 0.4366 | 0.4172 | 0.4307 | **0.4697** | 0.4550 |
| $d = 782585$ | P@3 ($B = 5000$) | **0.4018** | 0.3783 | 0.3866 | 0.4008 | 0.3870 |
| $K = 205443$ | P@5 ($B = 5000$) | **0.3816** | 0.3558 | 0.3619 | 0.3663 | 0.3550 |
| Amazon-670K | P@1($B = 10000$) | 0.4141 | 0.3946 | 0.3699 | **0.4489** | 0.4470 |
| $d = 135909$ | P@3($B = 10000$) | **0.3971** | 0.3581 | 0.3328 | **0.3980** | **0.3970** |
| $K = 670091$ | P@5($B = 10000$) | **0.3632** | 0.3305 | 0.3053 | 0.3600 | 0.3610 |

Table 3: Comparison of MACH and popular extreme classification algorithms on few public datasets. We notice that MACH mostly preserves the precision and slightly betters the best algorithms on half of the cases. These numbers also establish the limitations of pure tree based approaches FastXML and PfastreXML. Every 3 rows correspond to one dataset (color coded).

# E    Metrics of Interest

## E.1    Matching Metrics

- **Recall**@$K$: Recall is given by

$$\frac{|purchased\_products| \cap |topK\ predictions|}{|purchased\_products|}$$

- **MAP**@$k$: As name suggests, Mean Average Precision(MAP) is the mean of average precision for each query where the average precision is given by

$$sum\ i = 1 : K\ of\ (P@i * 1/K\ if predicition_i\ is\ a\ purchase)$$

- **MRR**@$k$: Mean Reciprocal Rank(MRR) is the mean of the rank of the most relevant document in the predicted list, i.e.,

$$\frac{1}{|Q|} \sum_{i=1}^{|Q|} \frac{1}{rank_i}$$

    where $rank_i$ is the position of the most purchased document for $i^{th}$ query.

- **nDCG**@$k$: Normalized Discounted Cumulative Gain(NDCG) is given by $\frac{DCG_K}{IDCG_K}$ where

$$DCG_K = \sum_{i=1}^{K} \frac{2^{rel_i} - 1}{\log_2(i + 1)}$$

$$IDCG_K = max_{i \in \{1,2,...,K\}} \frac{2^{rel_i} - 1}{\log_2(i + 1)}$$

    Here, $rel_i$ is 1 if prediction $i$ is a true purchase and 0 otherwise. $\log_2$ is sometime replaced with natural log. Either way, the higher the metric, the better it the model.

## E.2    Ranking Metrics

In addition to the above mentioned metrics, we care about Precision@1 in ranking. We want our top prediction to actually be purchased. Hence we would like to evaluate both P@1, P@1_weighted (weight comes from num_sessions like in the case of Matching). Further, for an ablation study, we also want to check if our top prediction is the top purchased document for a given query (we limit the purchases per query to just the most purchased document). Hence we additionally evaluate P@1_most_rel and P@1_most_rel_weighted .