[Reviews · NeurIPS 2019]

Reviewer 1



Quality: The paper is technically sound. The claims and proofs are well written and supported by experiments from both public and private datasets. This is a complete piece of the work. Clarity: The paper is mostly clearly written and well organized. However, the task goal discussion and the result part could be improved. Originality: The paper brought a new method to solve an existing task. The related works are clearly cited. The method is novel and different from past works. Significance: The result is significant and could become the new baseline in this area.

Reviewer 2



The paper presents a method for scaling up classifiers for tasks with extremely large number of classes, with memory requirements scaling with O(logK) for K classes. The proposed model is effectively using count-min sketch to transform a very large classification problem to a small number of classification tasks with a fixed small number of classes. Each of these models can be trained independently and in parallel. Experimental results on a number of multi-class and multi-label classification tasks shows that it either performs as well as other more resource-demanding approaches or it outperforms them, depending on the dataset, with a controlled reduction of model size based on the selection of the B and R parameters. When it is used for modeling retrieval as a classification with a very large number of labels, it outperforms Parabel, which is also limited by memory requirements.

Reviewer 3



This paper studies the task of extreme classification with a large amount of target categories. It developed a hashing-based algorithm, MACH. MACH leverage multiple hash functions that each maps the label classes into a small set of groups. Then a classifier is trained and applied for each hash mapping, on the reduced problem with much smaller amount of target classes. The prediction results of the sub-classifiers are then combined to re-constructed the final output. The proposed methods are demonstrated to be both efficient and effective in multiple datasets. Pros: a simple and efficient extreme classification method. The divide and concur approach works well on several datasets. Cons: The effectiveness of proposed method seems to be rather data dependent. It works well on ODP but not as much on ImageNet. There is no clear discussion of this divergence. The experimental set up and the motivation of 50M IR experiment are a little supervising. The standard IR approach is not to treat each target document as a target class. It is unconventional to treat the product search task as a multi-class classification task. The application of extreme classification in this setting, which is the main motivation of this paper (as stated in the title and introduction), seems arbitrary. The baseline compared in the production search setting is also rather weak. Only simple embedding-based ones are compared.

[Author Response · NeurIPS 2019]

**To R1:** Thank you very much for the positive comments. We are delighted that you appreciate our work. We'll modify the intro to add some background about Information retrieval.

Our baselines differ for Multi-class vs Multi-label datasets. Hence, the tables may show different baselines each time. We'll organize the plots/tables separately for both sub-sections. Thank you for pointing this out.

Great suggestion on choice of $B/R$. In theory, since there is a direct connection between count-sketches and MACH, we can work that out. Using Cauchy-Schwarz and Markov's inequality, we can get an $\epsilon - \delta$ relation (accuracy-failure probability trade-off) between $R$ and $B$ which goes like: if $max\ p_i >= \alpha$, then $P(|\hat{p}_i - p_i| < \epsilon) > 1 - \delta$ implies that $RB = \frac{1-\alpha^2}{\delta\epsilon^2}$. Based on our tolerance to $\epsilon$ and $\delta$, and the ease of data classification (given by $\alpha$), we can get an estimate of $RB$. We'll include a discussion about this in the final version.

Thank you for pointing the typo in 'num_trees, n_epochs ...' legend. We've corrected it to 'Parabel, n_epochs ...'.

**To R2:** Thank you very much for the positive comments. We are delighted that you appreciate our work.

On a second thought we think that changing title to focus on method is a great suggestion. We'll modify the title (if the PC permits) to reflect our proposed method. We'll also modify the intro to better synergize Extreme Classification and Information Retrieval.

**To R3:** Thank you for the positive comments. We would make necessary changes in the motivation to reflect better synergy between our method MACH and the task of Information Retrieval. Please see the following clarifications:

- Motivation for IR experiment: Posing Information Retrieval as a classification task is not unconventional. The baseline Parabel compared in this paper is a 1-vs all classifier model with some partial tree structure. It has been deployed on Bing Search Engine and it works really well. Posing IR as classification is known in literature and is at least as old as 2008 Li, Burges and Wu (NIPS 2008) " McRank: Learning to Rank Using Multiple Classification and Gradient Boosting" where they showed that classification loss is naturally a good surrogate for ranking (upper bound for DCG).

- Baselines: We compared against two tried and tested baselines from real search engines. The first one, Parabel is discussed in point 1. The second embedding baseline is deployed on our collaborators search engine. Other publicly available Extreme Classification algorithms are inferior to Parabel (as we see the metrics on Extreme Classification Repository). As mentioned in line 265, we tried running the best publicly known embedding model AnnexML on the same dataset. We varied embedding dimension among 256,512, number of learners among 4,8,16 and number of nearest neighbors among 10,50,100. Even the smallest configuration trained for 5 days without any progress. As noted in line 268, another well performing model SLEEC has a MATLAB code that cannot scale beyond 1M classes (as seen on Extreme Classification Repository).

- ODP vs ImageNet: ImageNet has 22K classes most of which are closely related. There are 1000 standard ImageNet classes and 2-hop an 3-hop classes which are fine-grained versions of original 1000. Closely related classes like different types of birds are prone to spurious prediction probabilities in any general ML algorithm. Further, ImageNet has dense features where accuracy takes a hit with approximations. Also, the larger the number of classes, the more the gains with MACH. We'll add a short discussion about this ImageNet disparity in the paper. We omitted it due to space constraints.

[Meta-Review · NeurIPS 2019]

The paper presents a method for scaling up classifiers for tasks with extremely large number of classes, with memory requirements scaling with O(logK) for K classes. The proposed model is uses count-min sketch to transform a very large classification problem to a small number of classification tasks with a fixed small number of classes. Each of these models can be trained independently and in parallel. Experimental results on a number of multi-class and multi-label classification tasks shows that it either performs as well as other more resource-demanding approaches or it outperforms them, The methodological contribution is significant and it would be the baseline of future studies.